# Catabolic Reductive Dehalogenase Substrate Complex Structures Underpin Rational Repurposing of Substrate Scope

**DOI:** 10.3390/microorganisms8091344

**Published:** 2020-09-02

**Authors:** Tom Halliwell, Karl Fisher, Karl A. P. Payne, Stephen E. J. Rigby, David Leys

**Affiliations:** 1Manchester Institute of Biotechnology, University of Manchester, 131 Princess Street, Manchester M1 7DN, UK; Thomas.halliwell@manchester.ac.uk (T.H.); karl.fisher@manchester.ac.uk (K.F.); karl.payne@manchester.ac.uk (K.A.P.P.); steve.rigby4@btinternet.com (S.E.J.R.); 2Future Biomanufacturing Research Hub (FutureBRH), Manchester Institute of Biotechnology, University of Manchester, 131 Princess Street, Manchester M1 7DN, UK

**Keywords:** reductive dehalogenase, cobalamin, iron-sulphur clusters, bioremediation, EPR, X-ray crystallography

## Abstract

Reductive dehalogenases are responsible for the reductive cleavage of carbon-halogen bonds during organohalide respiration. A variety of mechanisms have been proposed for these cobalamin and [4Fe-4S] containing enzymes, including organocobalt, radical, or cobalt-halide adduct based catalysis. The latter was proposed for the oxygen-tolerant *Nitratireductor pacificus pht-3B* cataboli*c* reductive dehalogenase (NpRdhA). Here, we present the first substrate bound NpRdhA crystal structures, confirming a direct cobalt–halogen interaction is established and providing a rationale for substrate preference. Product formation is observed *in crystallo* due to X-ray photoreduction. Protein engineering enables rational alteration of substrate preference, providing a future blue print for the application of this and related enzymes in bioremediation.

## 1. Introduction

The reductive dehalogenases (RDases) are key enzymes in organohalide respiration, which is performed by a unique subset of bacteria [1,2]. The RDases are cobalamin-dependent and able to selectively cleave carbon-halogen bonds through reduction. As a result, RDase containing organisms have been targeted as a potential bioremediation tool for the clean-up of contaminated anaerobic sites where organohalides have accumulated, often due to improper disposal [3,4,5,6].

The RDases can be split into two classes: (i) the canonical respiratory dehalogenases, which use a halogenated compound as a final electron acceptor during organohalide respiration and (ii) the catabolic respiratory dehalogenases that occur in the catabolic pathways of non-organohalide respiring bacteria [7,8,9]. The oxygen-sensitive respiratory RDases contain a twin arginine-translocation signal (TAT), allowing transport across the periplasmic membrane followed by association with a membrane anchor RdhB [10]. In contrast, the catabolic RDase enzymes lack the TAT signal and tend to be oxygen-tolerant, suggesting these might provide robust future bioremediation catalysts [7,10,11,12,13].

Structural insight into RDase function is based on two enzymes that have been crystallised, the respiratory perchloroethene (PCE) *Sulfurospirillum multivorans* reductive dehalogenase (PceA) [14,15], and the catabolic reductive dehalogenase from *Nitratireductor pacificus pht-3B* (NpRdhA) [16]. Both structures show remarkable similarities, despite belonging to a different class of dehalogenase and having distinct substrate specificity (Figure 1A) [10].

Both enzymes contain a conserved redox chain consisting of cobalamin (norpseudo-cobalamin in the case of PceA) and 2 [4Fe-4S] clusters. Cobalamin is located deep within the enzyme core while the 2 [4Fe-4S] clusters are positioned close to the surface of the protein, with one cluster within Van der Waals distance to (norpseudo-)cobalamin, ensuring efficient electron transfer [14]. PCE bound protein crystals show placement of substrate in the proximity of the cobalamin and a number of conserved residues [14].

Reductive dehalogenation requires two electron transfers and proton donation [10,14] and has been proposed to occur via organocobalt adduct or radical mechanisms (recent mechanistic proposals summarized in Figure 1B) [17,18]. However, the structure of the PceA and NpRdhA active sites were found to be incompatible with an organocobalt adduct, due to direct clashes between the protein and the proposed adduct [14].

Recent papers have shown that PceA mediated dehalogenation is not limited to small poly-brominated or chlorinated ethenes, but also occurs with poly-brominated and chlorinated phenols [19]. EPR studies, and analysis of PceA crystals in complex with brominated-phenols did not detect any direct cobalt-substrate interaction, leading to the suggestion the enzyme operates via long range electron transfer from cob(I)alamin to substrate [15].

In contrast, no ligand complexes are available for NpRdhA. While EPR-spectroscopic analysis of NpRdhA in the absence of a halogen shows a typical spectrum of cobalamin in a base-off form (with water occupying the β-axial ligand position), in the presence of a halogenated substrate, the EPR spectrum shows (super)hyperfine features resulting from interaction of the cob(II)alamin species with the halide atom [16]. In fact, a chlorine atom is proposed to be the β-axial ligand of cobalamin in the NpRdhA structure. These observations, underpinned by density functional theory calculations on a docked 3,5-dibromo-4-hydroxybenzoic acid NpRdhA substrate complex, led to the proposal of a cobalt-halide adduct mechanism, based on cob(I)alamin nucleophilic attack on the halogen (Figure 1B).

With both crystal structures apparently ruling out formation of an organo-cobalt adduct, it remains unclear to what extent the respective radical and cobalt-halide mechanism account for the diverse reactions catalysed by the wider RDase family [20]. However, mechanistic evaluations of RDases using DFT calculations have suggested a requirement for “intimate contact” between substrate halogen and cobalamin cobalt, allowing orbital overlapping and inner sphere electron transfer [20,21,22].

In this paper, we present an improved NpRdhA crystallisation method that has allowed determination of a number of NpRdhA substrate bound structures. These confirm the presence of a cobalt-halide substrate interaction and illuminate substrate preference in this enzyme. We also demonstrate that X-ray photoreduction driven *in crystallo* catalysis occurs, and that rational protein engineering allows modest alteration in substrate preference.

## 2. Materials and Methods

### 2.1. Molecular Biology

The *N. pacificus* pht-3B reductive dehalogenase (NpRdhA, WP_008597722.1) was cloned for the production of protein containing a N-terminal hexahistidine tag. The vector pPT7 was used for expression of NpRdhA in *B. megaterium* allowing protein induction via the xylose promoter. NpRdhA inserts were PCR amplified using the appropriate primers and the CloneAmp HiFi PCR Premix (Takara Bio Europe SAS, Saint-Germain-en-Laye, France). PCR products were cloned into the desired vector using Infusion HD enzyme mix (Clontech, Takara Bio Europe SAS, France). NpRdhA variants were created using QuikChange^TM^ mutagenesis primers using the wild type N-terminal NpRdhA containing pPT7 as template. DNA template was removed by DpnI digestion (10 U, 1 h, 37 °C, N.E.B, Ipswich, MA, USA) and clean up performed (Macherey Nagel NucleoSpin^®^ Gel and PCR Clean-up kit). The PCR products were transformed into Stella competent cells and the presence of the desired mutation confirmed by DNA sequencing before the purified plasmid underwent transformation into the desired host strain. All primers used in this study can be seen in Table 1.

### 2.2. Transformation Protocols

Mineral media *B. megaterium* protoplast transformations were performed as described in [23].

### 2.3. Heterologous Expression

*B. megaterium* was grown in a Type NLF 22, 30 L BioEngineering fermenter containing 22 L Terrific Broth (Formedium, Kings Lynn, UK) at 37 °C. Once at an OD_600_ of 1, the temperature was reduced to 18 °C and the media supplemented with 1 μM vitamin B12 and 50 μM ammonium iron (II) sulphate for overnight protein induction. Protein expression was induced by addition of 0.1% xylose.

### 2.4. Protein Purification

NpRdhA containing cell pellets were re-suspended in lysis buffer (50 mM Tris pH 7.5, 200 mM NaCl) with DNase, RNase (Sigma, Gillingham, UK), and EDTA-free protease inhibitor tablets (Roche, Welwyn Garden City, UK). Cells were lysed using a cell disruptor (Constant Cell Disruption Systems, Daventry, UK) at 20 Kpsi and lysate clarified by centrifugation in a Beckman Coulter Optima L-100 XP ultracentrifuge at 185,000× *g* for 1 h at 4 °C. NpRdhA clarified lysate was applied to a 10 mL Ni-NTA agarose drip column (Qiagen, Manchester, UK) pre-equilibrated with lysis buffer at 4 °C. The column then underwent successive column washes with lysis buffer containing 15 mM and 30 mM imidazole (4 CV of each) and protein was eluted with 20 mL of 250 mM imidazole. Fractions containing NpRdhA were collected and concentrated using a 30 kDa molecular weight cut-off Vivaspin (GE Healthcare, Amersham, UK) spin column centrifugal concentrator, in a Sigma 3-16PK centrifuge fitted with an 11,180 rotor at 3894× *g* and 4 °C. Imidazole was removed using a 10 mL Centripure P100 Zetadex gel filtration column (EMP Biotech, Berlin Germany). NpRdhA was loaded according to the manufacturer’s instructions and eluted into 50 mM Tris pH 7.5, 200 mM NaCl. Purified protein concentration was estimated for NpRdhA by UV-visible absorbance spectroscopy using a Cary 50 UV-Vis spectrophotometer (Agilent, Stockport, UK), using the extinction coefficients ε_280_= 77,810 M^−1^ cm^−1^ (calculated from the primary amino acid sequence using the ProtParam program on the ExPASy proteomics server). Protein purity was judged by applying samples to a Bio-Rad (Watford, UK), Mini-Protean TGX stain free precast SDS-gel with a 4–20% gradient and visualized with a Bio-Rad Gel Doc^TM^ EZ Imager. 

### 2.5. Cofactor Analysis

The iron content of NpRdhA was confirmed colorimetrically with bathophenanthroline after acid denaturation. Iron was extracted from the protein by adding an equal volume of 2 M HCl and heat denaturing at 80 °C for 10 min. After removal of precipitate by centrifugation, a suitable quantity of the sample (10–200 μL) was taken for assay with bathophenanthroline using the method previously described [16] before measurement of the absorbance at 535 nm. Iron concentrations were determined from an iron standard curve over the range 0–50 nmol.

Cobalamin concentration was estimated after extraction by dicyano-complex formation. Cyanide extraction involved mixing 0.4 mM protein and potassium cyanide (10 mM) followed by heating of the sample in a fume hood for 20 min at 80 °C. Measurement of the cyanocobalamin UV-visible spectrum and quantification of concentration using the 550 nm reading (ε = 8.7 mM^−1^ cm^−1^) allowed determination of the protein bound cobalamin concentration.

### 2.6. NpRdhA Activity Analysis

Activity analysis was performed using either a spectrophotometric activity assay using reduced methyl viologen as an artificial electron donor or using a non-cognate spinach ferredoxin and *E. coli* flavodoxin reductase system for delivery of electrons from NADPH [24]. Both reactions were carried out under anaerobic conditions and contained 1 μM NpRdhA. 

Methyl viologen reactions were performed at 20 °C and contained 150 μM dithionite reduced methyl viologen and the desired concentration of substrate (3,5-dibromo-4-hydroxy benzoic acid) and reactions started by the addition of NpRdhA followed by mixing by inversion. The NpRdhA dependent rate of methyl viologen oxidation was measured at 578 nm (ε = 9.78 mM^−1^ cm^−1^) using a Cary UV-Vis spectrophotometer over 5 min and fitted to a linear rate where appropriate. 

Reactions using the non-cognate system contained 5 mM substrate, 10 mM NADPH, 100 μM spinach ferredoxin and 10 μM *E. coli* flavodoxin reductase and were started by addition of NpRdhA. Reactions were sealed within 2 mL amber crimp-top HPLC vials and incubated for 30 min at 30 °C. Reactions were stopped by the addition of 6% final concentration trichloroacetic acid, centrifuged at 141,00× *g* to remove precipitated protein and analysed by HPLC.

All activity measurements were performed in triplicate and are reported as mean ± standard deviation. Statistical analysis was performed in Prism 8 [25] using the k_cat_ model.

### 2.7. HPLC Analysis

HPLC analysis was performed on an Agilent 1260 Infinity HPLC with a UV diode array detector attached. The stationary phase used was a Kinetex® 5μ C18 100 Å column, 250 × 4.6 mm. The mobile phase was water/acetonitrile (50:50) containing 0.1% trifluoroacetic acid at flow rate of 1 mL min^−1^ for 10 min. 

### 2.8. Crystallogenesis

Crystallogenesis was performed using the Manchester Protein Structure Facility. The protein of interest was concentrated using a 30 kDa molecular weight cut-off Vivaspin centrifugal concentrator (Sartorius AG, Goettingen, Germany) until at a final concentration of 30 mg/mL. Protein crystallization was performed using the sitting drop technique in Swissci crystal trays. Crystal trials were set up in various conditions using a mosquito^®^ LCP robot (TTP Labtech, Cambridge, UK) to mix 200 nL mother liquor with 200 nL sample (concentrations between 7.5–30 mg/mL). Trays were incubated at 4 °C and the presence of crystals determined using microscopy.

### 2.9. Diffraction Data Collection and Data Processing 

Crystals were flash frozen in liquid nitrogen and supplemented with 10% PEG 200 prior to freezing. Crystal soaking was performed by submersion of the crystal in 100 mM of the desired substrate (in mother liquor) before freezing in liquid nitrogen. 

Diffraction data were collected at 100 K using the Diamond beamlines. Collected data were processed using the CCP4 molecular graphics suite [26] and Phenix [27]. Molecular replacement was performed using Phaser-MR [28] using the PDB: 4RAS as a model. Refinement was performed using Phenix.refine [27] and manual rebuilding was performed in Coot [29].

### 2.10. EPR Spectroscopy

Samples of 100 μM NpRdhA altered proteins were prepared as isolated or reduced and transferred in volumes of 300 μL into 4 mm Suprasil quartz EPR tubes (Wilmad, Vineland, NJ, USA) under anaerobic conditions. The tubes were anaerobically sealed, frozen and stored in liquid nitrogen. Experimental parameters were as follows: microwave power 0.5 mW, modulation frequency 100 kHz, modulation amplitude 5 G, temperature 30 K. Spectra were obtained using a Bruker ELEXSYS E500 spectrometer (Coventry, UK), Super high Q resonator (ER4122SHQ), Oxford Instruments (Abingdon, UK) ESR900 cryostat and ITC503 temperature controller. Spectra were sums of between 8 and 16 scans.

## 3. Results

### 3.1. Purification and Characterization of N-Terminally Hexa-Histidine Tagged NpRdhA 

Heterologous expression and purification of N-terminally tagged NpRdhA (NpRdhA^N^) was performed in *B. megaterium* as previously described for the C-terminal tagged NpRdhA (NpRdhA^C^) [16]. Elution fractions from Ni-NTA agarose purification were brown in colour and the corresponding UV-visible spectrum fractions contained broad features at 420 nm (indicative of the presence of the [4Fe-4S] clusters), and a shoulder at approximately 310 nm indicating the presence of cob(II)alamin. 

Relocation of the hexa-histidine tag in NpRdhA^N^ did not affect cofactor incorporation when compared to purified NpRdhA^C^ (Figure 2), with similar cobalamin content (29.1 ± 1.5% and 33.6 ± 3.1%, respectively) and [4Fe-4S] cluster levels (6.3 ± 0.2 and 6.5 ± 0.1, respectively). Furthermore, a comparable 3,5-dibromo-4-hydroxybenzoic (35-DB-4-OH) reductase activity was obtained (736 ± 66 and 690 ± 34 μM product min^−1^ mg ^−1^ for NpRdhA^C^ and NpRdhA^N^ respectively, using reduced methyl-viologen as an artificial electron donor).

### 3.2. High-Resolution Crystal Structure of N-Terminally Tagged NpRdhA

NpRdhA^N^ crystal trials were performed using protein concentrations of 10, 15 and 20 mg/mL. Crystals were obtained in 0.1 M Bis-Tris propane (pH 8.5), 0.2 M sodium nitrate and 20% *w/v* PEG3350, all with a regular rhombic shape and brown in colour. Crystals belonged to the same C2 space group as NpRdhA^C^ crystals, but yielded a maximum resolution of 1.7Å, with a majority of the data sets diffracting at ~ 2.0 Å (as opposed to NpRdhA^C^ crystals rarely diffracting better than 3.0 Å). This represents an improved and more reproducible source of crystals than obtained previously [16]. 

The structure of NpRdhA^N^ (1.7 Å) was solved via molecular replacement using the NpRdhA^C^ structure (PDB code: 4RAS) as a model (Table 2) [27]. The asymmetric unit (AU) contains three monomers that are very similar, with a pairwise alignment rmsd of 0.115–0.148 Å. An overlay of the crystal packing of both NpRdhA^C^ and NpRdhA^N^ reveals the relative position of molecules B and C is particularly affected by the distinct position of the affinity tag, the direct consequence being reduced mobility for these molecules and associated improved diffraction in the case of NpRdhA^N^. 

A comparison of the NpRdhA^C^ and NpRdhA^N^ structures by alignment of respective chain A monomers yields an rmsd of 0.148 Å over 614 residues, confirming the distinct position of the hexa-histidine tag has little effect on the protein structure. Furthermore, alignment of active site residues from NpRdhA^C^ and NpRdhA^N^ show excellent agreement with an rmsd of 0.092 Å for the 10 residues selected. A chloride ligand, likely derived from the purification buffer, is modelled as the β–axial ligand at a cobalt-chloride distance of 2.6 Å (Figure 3). 

### 3.3. Crystal Structure of NpRdhA^N^ Ligand Complexes

Substrate soaking of NpRdhA^N^ crystals was performed using a number of organohalide substrates [16]. Diffraction data were collected for crystals soaked with 3,5-dibromo-4-hydroxybenzoic acid (35-DB-4-OH, 1.99 Å) and 3-bromo-4-hydroxybenzoic acid (3-B-4-OH, 1.99 Å). A difference Fourier transform using the high-resolution NpRdhA^N^ model revealed extra electron density in the active site region (Figure 4), showing clear density for 35-DB-4-OH, as well as electron density corresponding to multiple conformations of 3-B-4-OH. 

The modelling of 35-DB-4-OH into the additional electron density observed at the active site of NpRdhA was performed using ligand fit [30]. Initially, two molecules of 35-DB-4-OH were found respectively for chain A and chain B, each with overall 0.84 correlation coefficient (CC). The 3rd molecule was manually fitted into the extra density seen at the chain C active site, and all 35-DB-4-OH bind in a similar orientation. However, careful analysis of the difference electron density corresponding to the Br atoms revealed that this was not fully consistent with presence of the 35-DB-4-OH substrate only. X-ray mediated photoreduction readily occurs at synchrotron sources [30,31], and this could drive reductive dehalogenation *in crystallo*. Following the modelling of only partial (i.e., 40%) 35-DB-4-OH occupancy, residual electron density is consistent with the dehalogenation product (3-B-4-OH) and a Co(II) ligating Br ion (both modelled at 30% occupancy).

35-DB-4-OH is located above the cobalamin (Figure 4A,B) oriented so that one bromide occupies the β-axial ligand position, with a Co-Br distance of 3.2 Å. Little evidence of substrate induced conformational changes can be observed from a comparison with the substrate free NpRdhA^N^ structure, with only minor reorientations of N452 and F291 that accommodate ligand binding. The active site of NpRdhA^N^ can be divided into two regions; the first contains a number of hydrophobic residues (F290, F310 and P311, A419 and M423) responsible for occluding solvent and orientation of the substrate. The second region is polar in nature and contains residues responsible for the binding of the 35-DB-4-OH hydroxyl group and putative proton transfer during catalysis. Residues K488, R552 and S422 are within hydrogen bonding distance of the 35-DB-4-OH hydroxyl group (2.8, 2.7 and 3.3 Å, respectively, Figure 4A). Previous studies revealed severely compromised activity for K488Q and R552L variants, whereas the S422M mutation resulted in insoluble protein [16,24]. 

The conserved residues Y426 and K488 have previously been implicated as part of a catalytic dyad acting as the key proton donor. Residue Y426 is oriented pointing towards the cobalamin cobalt and the ligating bromine, with a closest Y426-Br distance of 3.2 Å. The Y426 phenolic oxygen is located 3.5 Å from the halogenated carbon atom, in an ideal position for proton donation (Figure 4B). 

The position of the 3-B-4-OH product formed *in crystallo* occupies the same active site region, but the molecule has moved approximately 0.8 Å away from the cobalamin plane, increasing the distance between the reduced carbon and the bromide ion to 3.1 Å. A strong hydrogen bond with S422 is formed as a consequence (distance decreased from substrate 3.3 Å to product 2.5 Å). The bromide ion in turn is positioned at 2.4 Å from the central cobalt in the β-axial ligand position (Figure 4C).

A comparison with previously reported DFT models of the 35-DB-4-OH and 3-B-4-OH NpRdhA complexes [20] reveals qualitatively similar substrate/product orientation to those observed in the crystal structures (Figure 4D). Several side chains in the DFT model adopt different conformations, presumably a consequence of the lack of stabilising packing interactions due to the limited size of the model used. 

The structure of the NpRdhA^N^ protein in complex with 3-B-4-OH obtained through soaking with 3-B-4-OH is shown in Figure 4E–F. The additional electron density in the active site region corresponding to 3-B-4-OH can be modelled by three distinct conformations, again guided by the higher electron density of the single bromine atom. Only one conformation provides a halogen beta axial ligand and thus resembles the 35-DB-4-OH substrate complex (occupancy 30%).

One of the two distinct, non-Co-ligating conformations corresponds to that of the previously observed 35-DB-4-OH *in crystallo* reduction product (occupancy 30%), while the other occupies the entrance of the active site cavity (occupancy 40%). In the latter case, the 3-B-4-OH benzoic acid group forms a salt bridge with K307 (3.1 Å) while the phenolic oxygen interacts with Y426 (2.5 Å). 

The 3-B-4-OH reductase activity of NpRdhA has previously been shown to be significantly lower (~20%) compared with the 35-DB-4-OH reductase activity [16,24]. The 3-B-4-OH bound structure reveals this is likely due to multiple conformations, only one of which appears catalytically competent. Indeed, inhibition of NpRdhA reductase activity due to inhibitory substrate binding modes likely accounts for the higher K_m_ and lower k_cat_ seen for this substrate in comparison to 35-DB-4-OH [32]. 

### 3.4. Structure of NpRdhA^N^ K488Q Variant

The NpRdhA variant K488Q was previously reported to lack 35-DB-4-OH reductase activity [16]. Crystals of the K488Q variant were obtained in the same condition as the wild type NpRdhA^N^ crystals. A representative electron density map for NpRdhA^N^-K488Q (2.1 Å) can be seen in Figure 5A. 

Alignment of chain A of K488Q and the NpRdhA^N^ structure gave an rmsd of 0.179 over 680 residues. The only major difference occurred, as expected, at position Q488. An overlay of the three K488Q NpRdhA monomers reveals multiple conformations of Q488 and adjacent residues S422 and R552, also implicated in substrate binding (Figure 5B).

Characterization of the K488Q^N^ cobalamin using EPR produces an “as isolated” protein spectrum (Figure 5D) quite similar to that of the wild type NpRdhA^C^ (Figure 5C) in the absence of chloride with g_┴_ = 2.34, A_┴_ = 72 G, g_||_ = 2.00, A_||_ = 146 G, indicating base off five coordinate cob(II)alamin with water as the fifth ligand to the cobalt ion, despite the presence of chloride ions in the buffer. The addition of excess substrate sharpens the spectrum (lines become narrower) but does not change the spectroscopic parameters (Figure 5E). Both of these observations stand in stark contrast to the wild type enzyme that exhibits superhyperfine coupling (A_||_ = 25.4 G) to chloride ion in chloride containing buffers [6] and a distinct change in both g and hyperfine coupling values on substrate binding (Figure 5G). Including the K488Q mutant in a 60 min incubation with excess substrate and a non-cognate redox system (consisting of *E. coli* flavodoxin reductase (EcFldR) and spinach ferredoxin (SpFdx) [32]) to support the transfer of electrons from NADPH gives rise to the same cob(II)alamin spectrum as the K488Q variant alone, plus a contribution from reduced SpFdx that is evidently not reoxidised by the K488Q (Figure 5F). Taken together, these results suggest that the K488Q mutant can bind substrate, but not productively and that the K488Q variant is incapable of accepting electrons from SpFdx. 

### 3.5. Rational Mutagenesis of NpRdhA

To further probe our understanding of substrate scope in NpRdhA, we explored whether rational protein engineering of the active site can be used to alter ligand specificity. Substrate bound NpRdhA^N^ structures show A419 as part of the pocket that accommodates the second i.e., non-ligating bromide atom of the 35-DB-4-OH substrate. We hypothesized that an A419M variant would render the enzyme unable to accommodate di-halogenated substrates, thus altering substrate specificity towards mono-halogenated substrates. Furthermore, A419M could prevent binding of mono-halogenated substrates in a product like conformation. Expression and purification of the C-terminal hexahistidine tagged A419M NpRdhA variant as described for NpRdhA^C^, resulted in pure protein of similar quality. Analysis of the reductase activity using a non-cognate redox system revealed the ability to dehalogenate 35-DB-4-OH, 3-B-4-OH, 35-DC-4-OH and 3-C-4-OH, at approximately 65%, 97%, 41% and 72% of the corresponding WT NpRdhA activity, respectively (Figure 6). Hence, the A419M mutation has indeed resulted in an enhanced preference for mono-halogenated phenolic substrates, although it did not increase the relative activity towards such compounds.

## 4. Discussion

The use of N-terminal hexa-histidine tagged NpRdhA has yielded improved crystals for this enzyme and allowed the determination of substrate bound NpRdhA and variant structures. The location of the tag does not affect NpRdhA cofactor content or 35-DB-4-OH reductase activity. However, the N-terminal NpRdhA version could be reproducibly crystallised and crystals obtained regularly diffracted to higher resolution than previously reported for the C-terminal tagged version [16]. This supports present and future studies aimed at elucidating ligand complexes and/or structure of NpRdhA variants.

The structure of the 35-DB-4-OH: NpRdhA complex reveals a network of polar interactions occurs between S422, K488 and R552 and the substrate hydroxyl group. In addition, the substrate interacts through hydrophobic interactions with the highly shape complementary active site, as well as through the Co(II)-Br ligation. By virtue of the high Br electron density, the presence of X-ray photoreduction induced 3-B-4-OH/Br^−^ products could also be observed, accounting for approximately 40% of the ligand density. This suggests that reductive cleavage of the Br-C bond is accompanied by proton transfer from Tyr426 concomitant with minor repositioning of Br- and 3-B-4-OH products in the active site. When soaking crystals with 3-B-4-OH, the latter is observed in multiple conformations, only one of which corresponds to the 35-DB-4-OH *in crystallo* reductive dehalogenation product observed. A second conformation, related by a 180 degree rotation, adopts a substrate-like conformation with the single bromine atom ligating the Co(II). The lower propensity of 3-B-4-OH to bind in a productive orientation might in part explain the lower activity seen with this substrate in comparison to 35-DB-4-OH [16].

In addition to ligand complex structures, the new crystallisation conditions also support structure determination of key variants. The K488Q mutation has previously been shown to abolish NpRdhA 35-DB-4-OH reductase activity [16]. The K488Q crystal structure confirms that no significant changes occur to the wider protein active site. EPR analysis of the K488Q variant reveals a sharpening of spectral features upon addition of substrate, but not the clear cobalt-halogen interaction seen with WT NpRdhA, suggesting non-productive binding with water remaining as the fifth ligand to the cobalt ion. Thus, it seems that the K488Q mutation disrupts the halide ion and substrate binding properties of the active site, likely a consequence of the disorder induced at positions S422 and R552, as well as perturbation of the Tyr426 pK_a_. Instead, we see a small but reproducible effect due to substrate binding at an undefined site normally masked in EPR studies of WT enzyme. It is possible that this site might correspond to the third 3-B-4-OH conformation observed near the entrance of the active site. The extent by which the K488Q mutation disrupts substrate binding casts light on the critical role K488 plays in this enzyme. Unfortunately, crystal structures of K488Q ligand complexes could not be obtained, despite several attempts. Interestingly, K488Q reduction could not be observed even in the presence of substrate, suggesting that electron transfer to NpRdhA might only occur when a *bona fide* Co-substrate halogen interaction has been established.

Construction of an A419M variant in an attempt to shift substrate specificity towards mono-halogenated substrates has met with some success. While the relative activity with 3-B-4-OH indeed exceeds that with 35-DB-4-OH for A419M, absolute values for 3-B-4-OH reductive dehalogenation have not improved. This suggests that the mutation either was unsuccessful in fully reducing conformational heterogeneity for 3-B-4-OH (in part supported by the fact 35-DB-4-OH activity was not zero) or that 3-B-4-OH reductive dehalogenation is not limited per se by the observed heterogeneity. A419M NpRdhA provides a further example of an RDase whose active site has been altered to affect substrate specificity of the enzyme [33]. This suggests NpRdhA and related enzymes can be considered a platform for future directed evolution to widen substrate of these enzymes for bioremediation purposes.

In view of the distinct observations made regarding the position of substrate and the lack of any direct halogen-cobalt interaction detection in the respiratory reductive dehalogenase PceA [14,15], further detailed studies are required to assess the diversity in exact mechanism employed to achieved carbon-halogen bond breakage (Figure 1). These ideally should focus on ensuring appropriate coverage of distinct reductive dehalogenases to cover a range of respiratory and catabolic enzymes reflecting the inherent diversity in terms of electron donor, substrate range, membrane-bound nature, and oxygen-sensitivity. 

## 5. Conclusions

The catabolic reductive dehalogenase NpRdhA was proposed to function via an unusual mechanism involving a cobalt-halide adduct [16,20]. Crystal structures of ligand complexes confirm that substrate:enzyme complex formation leads to a cobalt-halide adduct, placing a halogen of the bound substrate within coordination distance of the cobalamin cobalt. The Co(II)-substrate halogen complex is thus poised to receive an electron through the FeS clusters, and mutation of K488 abolishes the Co(II)-halogen interaction and any reductive dehalogenation activity. X-ray photoreduction leads to partial turnover *in crystallo* paving the way for future time-resolved crystallographic studies. Rational engineering attempts to affect substrate specificity have met with some success, supporting future studies exploring both mechanistic detail and substrate scope of this enzyme. Given the oxygen-tolerant nature of NpRdhA, the enzyme might thus provide an attractive target for bioremediation.

## Figures and Tables

**Figure 1 microorganisms-08-01344-f001:**
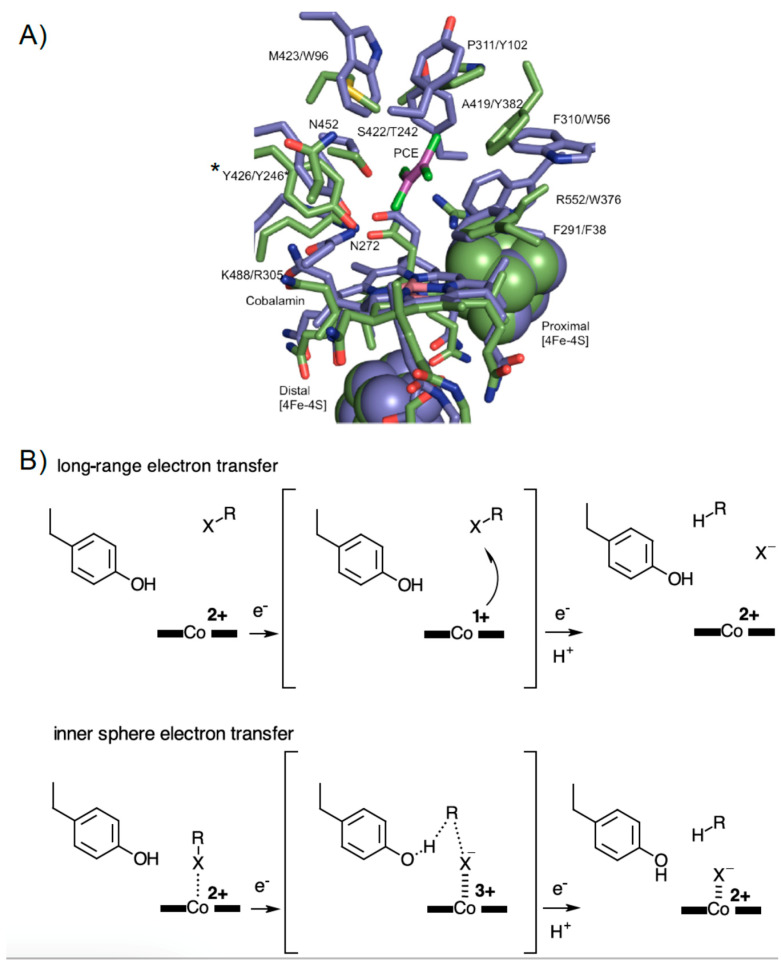
Reductive dehalogenase crystal structures and mechanistic proposals. (**A**) Overlay of the active site region of PceA (PDB: 4UR0, in blue) and NpRdhA (PDB: 4RAS, in green). Perchloroethene (PCE) is shown in magenta. PceA Tyr246 and the equivalent Tyr426 in NpRdhA (indicated by * are thought to function as a proton donor. Residue labels refer to NpRdhA/PceA, respectively. (**B**) An outline of the proposed long range versus short range/inner sphere mechanisms on the basis of substrate/ligand positions (inferred in case of NpRdhA) with respect to the cob(II)alamin. In each case, a range of possibilities with respect to the exact order of (or coupling between) electron and proton transfer steps is possible. Besides the distinct need for appropriate positioning of the substrate, the mechanisms also differ in the requirement for formation of the supernucleophile Co(I) species.

**Figure 2 microorganisms-08-01344-f002:**
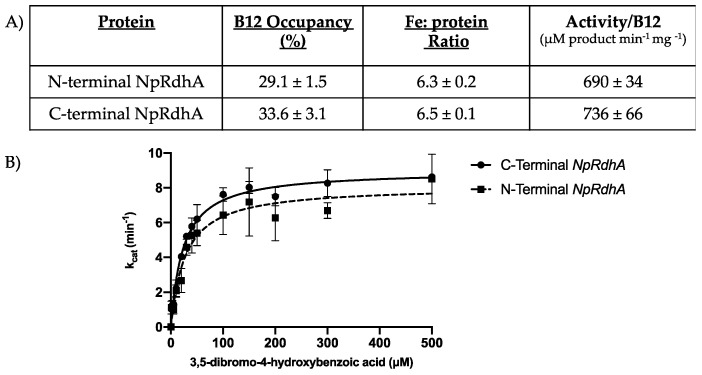
Comparison of N- and C-terminal His-tagged NpRdhA. (**A**) Table of cofactor content determined using cyanide extraction (B12) and bathophenanthroline (Iron). Activity/B12 was determined by measuring the dehalogenase activity with 3,5-dibromo-4-hydroxybenzoic acid and using reduced methyl viologen as an electron donor under an N_2_ atmosphere, with rates standardised to cobalamin incorporation levels; (**B**) Kinetic curve of NpRdhAs 3,5-dibromo-4-hydroxybenzoic acid reductase activity. Measurements were performed in triplicate and reported as mean ± standard deviation. Data was fit in Prism 8 [25] using the k_cat_ model.

**Figure 3 microorganisms-08-01344-f003:**
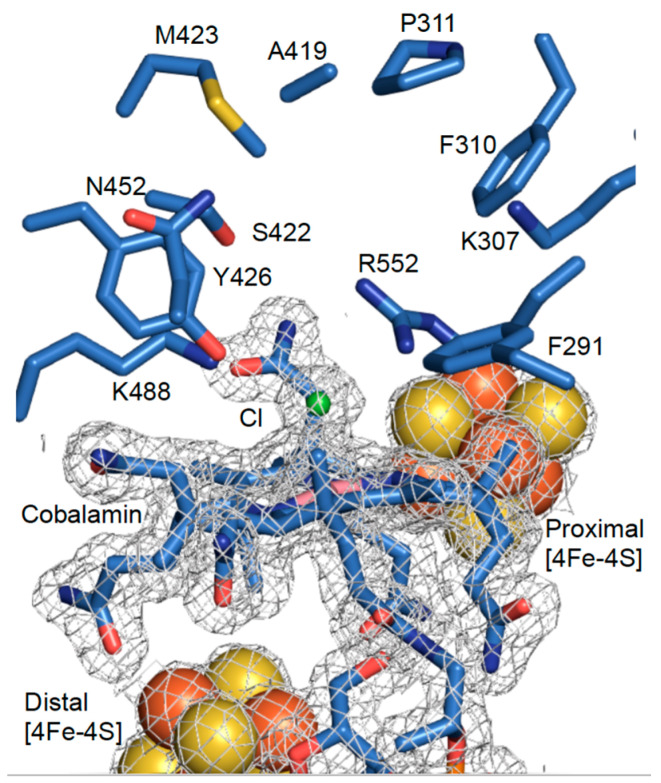
High resolution NpRdhA^N^ active site structure. Redox cofactors and active site of NpRdhA^N^ monomer A with corresponding electron density map contoured at 1σ (grey).

**Figure 4 microorganisms-08-01344-f004:**
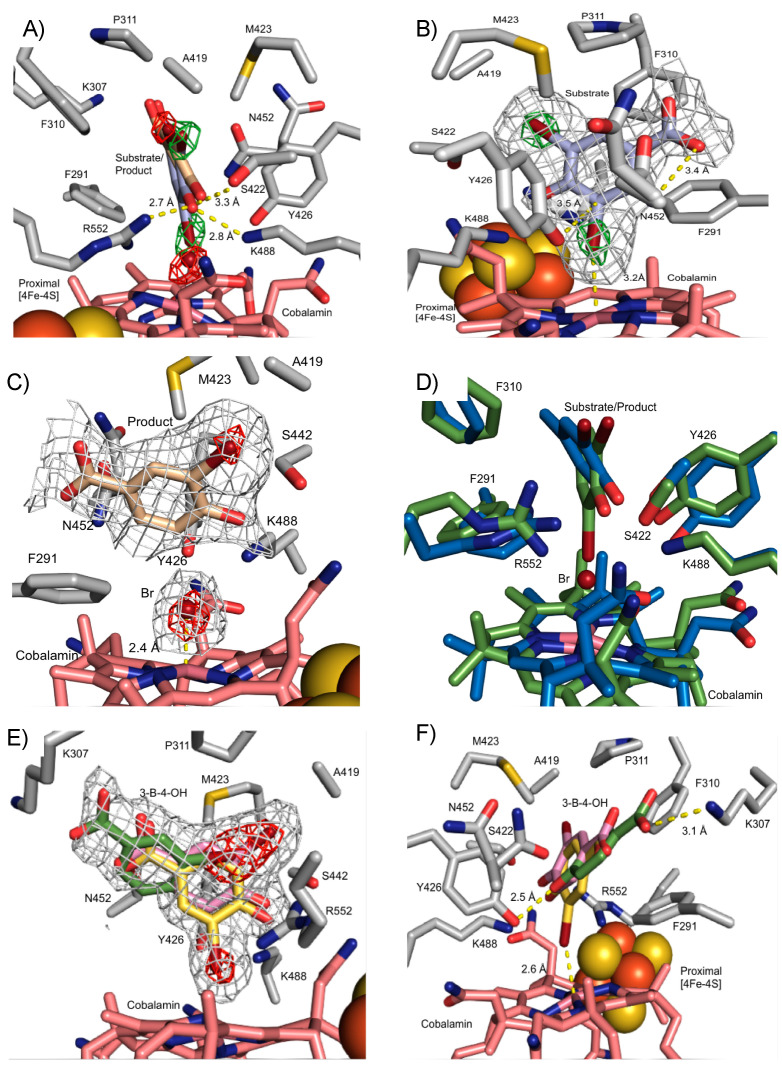
NpRdhA^N^ ligand complex structures. Grey mesh represents omit electron density polder maps to show ligand placement in NpRdhA^N^ contoured between 3–5σ. The position of bromides is represented by contouring polder maps between 5–11σ. (**A**) NpRdhA^N^ with 3,5-dibromo-4-hydroxybenzoic acid (35-DB-4OH, blue) substrate and 3-bromo-4-hydroxybenzoic acid (3-B-4-OH, brown) *in crystallo* product. Bromine atoms indicated by omit maps contoured at 10σ (green; substrate) and 6σ (red; product). (**B**) The 35-DB-4-OH: NpRdhA^N^ complex reveals placement directly above cobalamin within hydrogen bonding distance of key residues, ideally placed for proton donation by Y426. (**C**) 3-B-4-OH: NpRdhA^N^ complex obtained *in crystallo* as consequence of X-ray photoreduction; (**D**) an overlay of corresponding substrate (green) and product (blue) DFT models in similar orientation to panel A [21]; (**E**) 3-B-4-OH: NpRdhA^N^ complex obtained through soaking (similar view to (**C**). Distinct 3-B-4-OH conformations are shown, substrate-like ligating the Co (yellow) and non-catalytically competent product-conformation (pink) as well as a more distant position at the entrance of the active site (green). Electron density corresponding to bromine atoms in red; (**F**) alternative view of the 3-B-4-OH: NpRdhA^N^ complex obtained through soaking, similar in orientation to panel (**A**).

**Figure 5 microorganisms-08-01344-f005:**
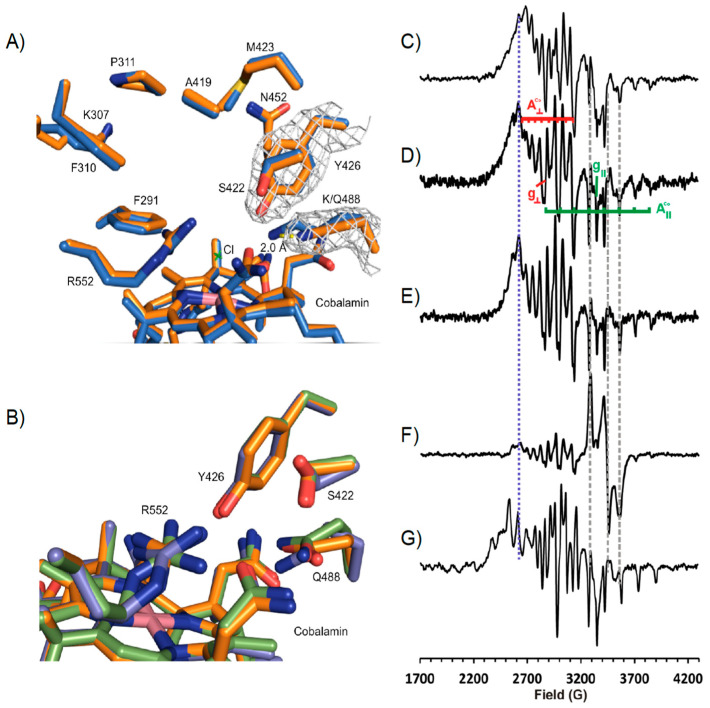
NpRdhA^N^-K488Q Structure and EPR Spectral analysis. (**A**) representative electron density for chain A Y426 contoured to 1σ, displayed in blue mesh. Maps generated using Fast Fourier Transform program in Coot. Alignment of wild type NpRdhA^N^ (blue) and K488Q (orange) resulted in an rmsd of 0.188 over 680 residues. Side chains only shown for simplicity. The Q488 amine group is removed 2.0 Å from the active site in comparison to K488. (**B**) Overlay of K488Q chains in asymmetric unit (rmsd 0.107–0.110) shows slight variations in hydroxyl binding residues positions. (**C**) X-band continuous wave EPR spectra of wild type NpRdhA alone; (**D**) K488Q NpRdhA alone; (**E**) K488Q NpRdhA + 20 mM 3,5-dichloro-4-hydroxybenzoic acid; (**F**) K488Q NpRdhA + 100 μM SpFdx + 100 μM EcFldR + 20 mM NADPH incubated for 60 min with 20 mM 3,5-dichloro-4-hydroxybenzoic acid; (**G**) WT NpRdhA + 20 mM 3,5-dichloro-4-hydroxybenzoic acid. Experimental parameters: microwave power 0.5 mW, modulation amplitude 5 G, spectra were sums of between 8 and 16 scans. All samples contained 100 μM NpRdhA and were measured at 30 K.

**Figure 6 microorganisms-08-01344-f006:**
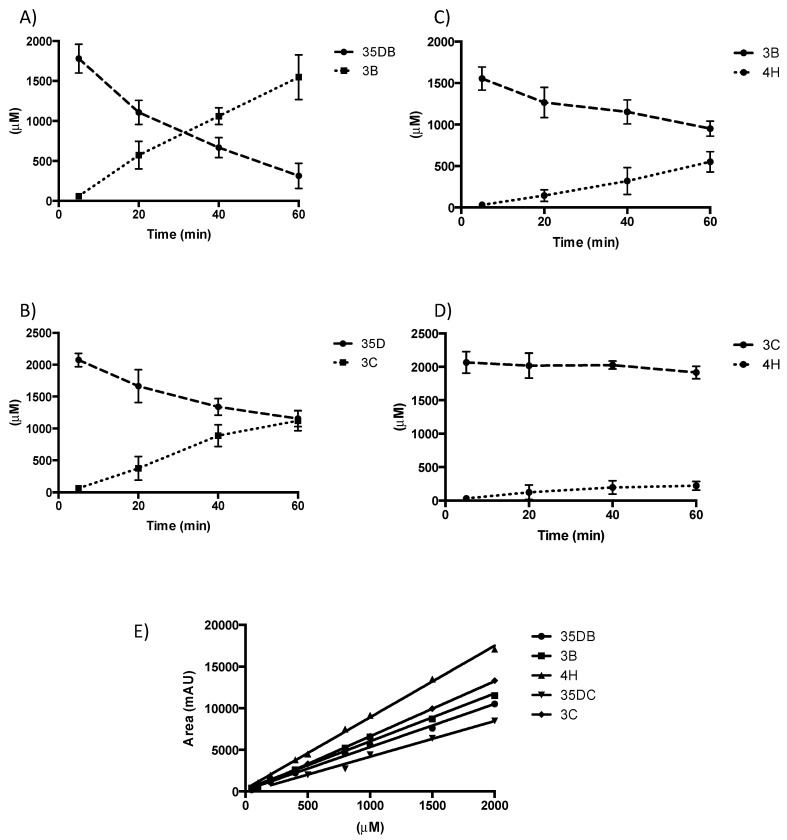
Rational substrate reprofiling through A419M mutation. HPLC product profiles obtained with NADPH and 1 μM A419M NpRdhA. (**A**) 3,5-dibromo-4-hydroxybenzoic acid (2 mM) rate = 76 μM/min; (**B**) 3-bromo-4-hydroxybenzoic acid (2 mM) rate = 15.4 μM/min; (**C**) 3,4-dichloro-4-hydroxybenzoic acid (2 mM) rate = 35.9 μM/min; (**D**) 3-chloro-4-hydroxybenzoic acid (2 mM) rate = 11.9 μM/min; (**E**) HPLC standardisation for each selected substrate – 35DB (3,5-dibromo-4-hydroxybenzoic acid), 3B (3-bromo-4-hydroxybenzoic acid), 4H (4-hydroxybenzoic acid), 35DC (3,5-dichloro-4-hydroxybenzoic acid), 3C (3-chloro-4-hydroxybenzoic acid).

**Table 1 microorganisms-08-01344-t001:** Primers used in this study.

Primer	Sequence
NpRdhA^N^-pPT7-F	GGAGGTGAAATGTACAATGGTCCAAACTAGTTTCGAACATCACCATCACCATCACTCAGCTGGTGAAAACTTATATTTCCAAGGCGCCCAGATCTCCATGCGCCTTTATTCCAATCG
NpRdhA^N^-pPT7-R	GGCCGGTACCGGATCCTTAGCCAGCGCTAGATTTAAGAC
NpRdhA-K488Q-F	TCCGTTTATTGGTCCTCGTTCCAAAAGTATCGTTTTTACAA
NpRdhA-K488Q-R	TTGTAAAAACATACTTTTGGAACGAGGACCAATAAACGGA
NpRdhA-A419M-F	AAGGAGCTTCAGGTGATGATTGGATTTCAATGAGTCAGTCCATGCGT
NpRdhA-A419M-R	ACGCATGGACTGACTCATTGAAATCCAATCAT CACCTGAAGCTCCTT

**Table 2 microorganisms-08-01344-t002:** Crystallographic data collection and refinement values.

Structure (PDB Code)	NpRdhA^N^ (6ZXU)	NpRdhA^N^:35-DB-4-OH (6ZXX)	NpRdhA^N^:3-B-4-OH (6ZY1)	NpRdhA^N^-K488Q (6ZY0)
Data Collection				
Space Group	C 2	C 2	C 2	C 2
*a, b, c* (Å)	175.11 170.96 107.87	180.18 170.39 107.96	175.87 170.10 107.65	176.35 169.57 107.80
β (°)	98.37	99.15	98.35	98.33
Resolution (Å)	121.7–1.73 (1.91–1.73)	123.1–1.99 (2.26–1.99)	106.51–1.99 (2.21–1.99)	106.66–2.13 (2.44–2.13)
R_meas_	0.104 (0.873)	0.149 (0.886)	0.181 (0.827)	0.196 (1.238)
I/σI	8.7 (1.6)	7.3 (1.8)	6.8 (1.7)	8.2 (1.8)
Completeness (%)	93.0 (58.0)	94.5 (75.5)	94.0 (78.5)	94.5 (74.6)
Redundancy	3.3 (2.3)	3.4 (3.2)	3.5 (3.1)	7.0 (6.7)
Refinement				
No. Reflections	201,352 (685)	97,499 (302)	112,203 (229)	84,097 (316)
R_work_/R_free_	0.1723/0.2026 (0.3239/0.3074)	0.1812/0.2233 (0.3106/0.3295)	0.1769/0.2180 (0.2354/0.2923)	0.1841/0.2328 (0.3054/0.5585)
Nr Atoms Protein	15,975	16,128	15,902	16,508
Nr Atoms Solvent	1703	696	968	305
Ion/Ligand	15	21	15	12
Protein B-factor (Å^2^)	27.1	31.7	23.7	36.0
Solvent B-factor (Å^2^)	33.3	32.9	27.8	34.2
Rmsd Bond Length (Å)	0.009	0.010	0.014	0.010
Rmsd Bond Angles (°)	1.556	1.507	1.072	1.541

Values for the highest resolution shell are shown in parenthesis.

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
