# Peer review of "Catabolic Reductive Dehalogenase Substrate Complex Structures Underpin Rational Repurposing of Substrate Scope"

_microorganisms, 2020, doi:10.3390/microorganisms8091344_

Round 1

Reviewer 1 Report

This paper from Tom Halliwell and colleagues reports the crystal structure of catabolic reductive dehalogenase from Nitratireductor pacificus pht-3B bound to 3,5-dibromo-4-hydroxybenzoic acid, a breakdown product of the herbicide bromoxynil.  This work is consistent with the authors' previous scientific production and represents an interesting evolution as they show for the first time a direct cobalt-halogen interaction. The article is well written, and the introduction provides adequate background.

The authors focused their attention on a reductive dehalogenase with an N-terminal tag, whose crystals show better diffraction than the C-terminal tagged one, without significantly altering the overall performance of the enzyme.  

Below, some minor issues that need to be addressed.

Figure 1, part B: the electrons balance around the cobalt atom is not represented clearly. The author should improve the graphics related to the number of electrons switched between the metal atom and the substrate.

Fig 2, part B: the quality of the figure needs to be improved. In addition, it requires a more detailed explanation of how the plot has been derived. At the Y axis the reaction rate is expressed as [min-1], rather than as the ratio between concentration and time. [min-1] would better fit a kinetic constant (kcat) if the reaction is of the first order.

Do the authors have data on the native protein without tags? Is it possible, in this specific case, to design a protein with a cleavable tag in order to fully characterize it in the native form?

The authors performed also protein engineering of the active site to alter ligand specificity.  They designed a A419M variant to evaluate the activity on mono-halogenated substrates. Why was this variant designed with the C-terminal tag considering that the study is set up on the N-terminal tagged enzyme?

They also designed a NpRdhAN  K488Q mutant that can bind substrate that cannot be converted   probably due to the lack of cobalt-halogen interaction

Taken together, these data show that this is an excellent system for designing reductive dehalogenase variants that may be increasingly suitable for environmental applications. Considering that organohalide pollution is a serious global environmental issue, reductive dehalogenases could indeed represent a promising natural tool for bioremediation. For this reason, I believe that this work is suitable for publication on Microorganisms after minor revision.

Author Response

Reviewer 1

  1. Figure 1, part B: the electrons balance around the cobalt atom is not represented clearly. The author should improve the graphics related to the number of electrons switched between the metal atom and the substrate.

We have updated the figure as requested.

  1. Fig 2, part B: the quality of the figure needs to be improved. In addition, it requires a more detailed explanation of how the plot has been derived. At the Y axis the reaction rate is expressed as [min-1], rather than as the ratio between concentration and time. [min-1] would better fit a kinetic constant (kcat) if the reaction is of the first order.

Figure 2 part B quality has been improved both in resolution of the image and presentation of the data itself, to make a more identifiable difference between the N and C-terminal NpRdhA data. The Y-axis has been changed to kcat (min-1) as the kcat model was used to fit this data in Prism. A paragraph has been added to 2.6 to reflect the statistical methods used in this study.

  1. Do the authors have data on the native protein without tags? Is it possible, in this specific case, to design a protein with a cleavable tag in order to fully characterize it in the native form?

We have not attempted to create NpRdhA constructs without an affinity tag or with a cleavable tag and therefore have no data on the untagged protein. Due to the relatively low expression of the reductive dehalogenases purifying an untagged protein to homogeneity would be very difficult. Although a limited number of vectors can be used for the expression of proteins in Bacillus megateriumdue to the xylose inducible promoter, we are aware there are vectors that would provide a cleavable tag but we believe that characterisation of an untagged native protein would not alter the results presented in this manuscript and therefore is not germane to this study.

The authors performed also protein engineering of the active site to alter ligand specificity.  They designed a A419M variant to evaluate the activity on mono-halogenated substrates. Why was this variant designed with the C-terminal tag considering that the study is set up on the N-terminal tagged enzyme?

Characterisation of the A419M mutant was performed using a C-terminal hexahistidine tag prior to obtaining the high resolution N-terminal tagged enzyme crystals. Due to the negligible difference between the C- and N-terminal tagged NpRdhA this data wasn’t repeated with NpRdhAN.

Reviewer 2 Report

Dear Editor,

I carefully read the manuscript by Halliwell et al., which is really well written and balanced in its parts.

My comments for the authors:

  • A paragraph regarding statistical analysis is missing. Authors should include the paragraph in the methods, and specify, for example, how the numerical variables were expressed in the manuscript (e.g. mean and standard deviation).
  • In the discussion, authors should refer and discuss the limitations of their study.
  • Some references are not conform to the Instructions for Authors. Please, revise them accordingly. 
  • Some references need to be replaced with newer articles. In particular, ref. 15 and 18.

Author Response

  1. A paragraph regarding statistical analysis is missing. Authors should include the paragraph in the methods, and specify, for example, how the numerical variables were expressed in the manuscript (e.g. mean and standard deviation).

A paragraph has been added in section 2.6 describing the statistical methods used and how the numerical variables were expressed in the manuscript. The legend in Fig 2 has been updated also to reflect these changes.

  1. In the discussion, authors should refer and discuss the limitations of their study.

We have added following statement at the end of our discussion:

“In view of the distinct observations made regarding the position of substrate and the lack of any direct halogen-cobalt interaction detection in the respiratory reductive dehalogenase PceA [14,15], further detailed studies are required to assess the diversity in exact mechanism employed to achieved carbon-halogen bond breakage (Fig 1). These ideally should focus ensuring appropriate coverage of distinct reductive dehalogenases to cover a range of respiratory and catabolic enzymes reflecting the inherent diversity in terms of electron donor, substrate range, membrane-bound nature and oxygen-sensitivity. “

  1. Some references are not conform to the Instructions for Authors. Please, revise them accordingly. 

References have been revised according to the instructions for authors, updating those that were missing journal names and issue numbers.

  1. Some references need to be replaced with newer articles. In particular, ref. 15 and 18.

Older references have been replaced by newer ones including both 15 and 18. 

Round 2

Reviewer 2 Report

Dear Editor,

I carefully read the revised version of the manuscript and I considered authors' replay to my concerns. I think that the new version of the paper is significantly improved rather than the previous one.